## Special feature reviews

evolution, genomics

adaptation, climate change, complex life cycle, fitness, genomics, pleiotropy

**Authors for correspondence:**
Molly A. Albecker
e-mail: malbecker@gmail.com
Laetitia G. E. Wilkins
e-mail: megaptera.helvetiae@gmail.com

†Co-first authors.
One paper of a special feature 'Evolution in Changing Seas'. Guest edited by Katie E. Lotterhos, Molly Albecker and Geoffrey Trussell.

# Does a complex life cycle affect adaptation to environmental change? Genome-informed insights for characterizing selection across complex life cycle

Molly A. Albecker[1,†], Laetitia G. E. Wilkins[2,†], Stacy A. Krueger-Hadfield[3], Samuel M. Bashevkin[4], Matthew W. Hahn[5], Matthew P. Hare[6], Holly K. Kindsvater[7], Mary A. Sewell[8], Katie E. Lotterhos[9] and Adam M. Reitzel[10]

[1]Department of Biology, Utah State University, Logan, UT 84321, USA
[2]Max Planck Institute for Marine Microbiology (MPIMM), Celsiusstrasse 1, 28209 Bremen, Germany
[3]Department of Biology, University of Alabama at Birmingham, 1300 University Blvd, Birmingham, AL 35294, USA
[4]Delta Science Program, Delta Stewardship Council, 715 P Street 15-300, Sacramento, CA 95814, USA
[5]Department of Biology and Department of Computer Science, Indiana University, 1001 E. 3rd St., Bloomington, IN 47405, USA
[6]Department of Natural Resources and the Environment, Cornell University, 205 Fernow Hall, Ithaca, NY 14853, USA
[7]Department of Fish and Wildlife Conservation, Virginia Polytechnic Institute and State University, Blacksburg, VA 24061, USA
[8]School of Biological Sciences, University of Auckland, Auckland 1010, New Zealand
[9]Northeastern University Marine Science Center, 430 Nahant Rd., Nahant, MA 01918, USA
[10]University of North Carolina at Charlotte, 9201 University City Blvd., Charlotte, NC 28223, USA

MAA, 0000-0002-5121-8101; LGEW, 0000-0003-3632-2063; SAK-H, 0000-0002-7324-7448;
SMB, 0000-0001-7406-7089; MWH, 0000-0002-5731-8808; MPH, 0000-0001-8569-8951;
HKK, 0000-0001-7580-4095; MAS, 0000-0002-1595-7951; KEL, 0000-0001-7529-2771;
AMR, 0000-0001-5734-7118

Complex life cycles, in which discrete life stages of the same organism differ in form or function and often occupy different ecological niches, are common in nature. Because stages share the same genome, selective effects on one stage may have cascading consequences through the entire life cycle. Theoretical and empirical studies have not yet generated clear predictions about how life cycle complexity will influence patterns of adaptation in response to rapidly changing environments or tested theoretical predictions for fitness trade-offs (or lack thereof) across life stages. We discuss complex life cycle evolution and outline three hypotheses—ontogenetic decoupling, antagonistic ontogenetic pleiotropy and synergistic ontogenetic pleiotropy—for how selection may operate on organisms with complex life cycles. We suggest a within-generation experimental design that promises significant insight into composite selection across life cycle stages. As part of this design, we conducted simulations to determine the power needed to detect selection across a life cycle using a population genetic framework. This analysis demonstrated that recently published studies reporting within-generation selection were underpowered to detect small allele frequency changes (approx. 0.1). The power analysis indicates challenging but attainable sampling requirements for many systems, though plants and marine invertebrates with high fecundity are excellent systems for exploring how organisms with complex life cycles may adapt to climate change.

# 1. Introduction

Across eukaryotes, there is a myriad of different life cycle types that can include complex morphological changes within a single ploidy stage (e.g. metamorphosis) or among ploidy stages (e.g. haplodiplontic life cycles) [1–4]. Because organisms with single ploidy stages can develop from the same genome (or from the same genes if the organism has stages that differ in ploidy), trade-offs among stages are expected to be common. As a consequence, an organism's adaptive potential will be strongly influenced by not only the performance of each independent stage [5,6] but also by how selection at each stage cascades through the entire life cycle [7–10]. The potential for life cycles to promote or constrain adaptation to changing environments remains a significant gap in predicting organisms' vulnerabilities to environmental change. This is particularly true among organisms with free-living stages that differ in ploidy (e.g. algae, ferns; [11]).

In this synthesis, we discuss the main hypotheses about how selection is predicted to alter evolutionary outcomes from environmental change in species with complex life cycles. We ask how environmental change may shift stage-specific selection in ways that can promote or constrain adaptation, outline a conceptual framework for hypothesis testing and interpretation using genomics, and suggest experimental designs to study selection across an individual life cycle. Our goal is to provide a generalized conceptual and novel experimental framework that can be used broadly across eukaryotes and importantly, is not restricted by taxon or by the specific details of the life cycle (box 1). These goals build upon [17], which considered similar questions from a quantitative genomic perspective. However, the maturation of genomic methods has generated powerful new opportunities to understand how selection acting within individual life stages affects adaptive potential in response to changing environments. Here, we explore the value of genomic analysis of single-generation artificial selection experiments because they can provide direct evidence of genetic correlations (pleiotropy) in terms of allele frequencies, potentially minimizing the confounding ambiguities generated from environmental effects.

# 2. Complex life cycles: an overview

The various approaches and experimental systems used to study life cycle diversity have muddled terminologies, which has impeded synthesis on how organisms progress from 'the fixed points of egg [to] corpse' [18]. Here, we view the entire life cycle as the central unit (*sensu* [19]) and basic characteristic of an organism [20] onto which we can map life-history traits or phenotypes. 'Fitness component' refers to any life history trait or allele that is correlated with within-stage fitness or total fitness when all other traits or alleles (respectively) are held steady [21]. 'Within-stage fitness' describes the effect of fitness components on survival or reproduction within an isolated stage, while 'total fitness' describes the cumulative effect of fitness components on survival or reproduction across all life stages [22]. To provide terminology that encapsulates eukaryotic diversity to base current and future discussions of selection across life stages, we have established a glossary in box 1.

Evolutionary modifications in development and environmental variation alter the expression of the genome at discrete temporal scales and produce life cycle diversity. Understanding the mechanisms responsible for generating this diversity connects multiple biological sub-disciplines, including eco-evolutionary dynamics (eco-evo) and evolutionary developmental biology (evo-devo). The eco-evo perspective has viewed complex life cycles as the products of selection on limited energy budgets, in which individuals must allocate a finite amount of energy among competing expenses related to maintenance (metabolism), growth and reproduction [23]. Models predict optimal trade-offs between pairs of life-history traits: current and future reproduction [24], offspring size and number [25,26], maturation and life-span [27], and mortality risk and growth potential [28,29]. As a result, metamorphosis is hypothesized to have evolved as a mechanism decoupling competing selection pressures across life stages, such that different stages can perform independently of each other in response to different selection pressures [30–33]. This concept, in which stages are disconnected through a discrete switch point, is called 'adaptive decoupling' [32]. On the other hand, evo-devo perspectives have long appreciated the interconnectedness of adaptations through ontogeny and how these may affect evolution [34–36]. Indeed, since development is an explicitly time-dependent process, the study of selection through an evo-devo lens includes not only the 'how' but also the 'when' (in ontogeny) an adaptation may affect fitness.

In nature, both pleiotropy (leading to interconnectedness among stages) and adaptive decoupling (leading to independence among stages) can impact how selection at one stage may affect performance of subsequent stages and total fitness. To best predict outcomes of environmental change, we must develop tools and approaches to study interactions between genotype, environment, and development time, which is crucial in quantifying composite selection across life cycles and testing the response in nature [37]. For a full review on concepts related to adaptive decoupling, genetic correlations (with respect to traits), pleiotropy, as well as quantitative genomic methods traditionally used to study these concepts, we refer the reader to [17].

Quantitative genetics has been used to address our understanding of the diversity of complex life cycles across eukaryotes in the past [17]. Moreover, empirical work on genetic correlations across stages has predominantly focused on one or two stages, with few studies quantifying selection across the entire life cycle (see box 2). In these studies, the response is usually measured as a phenotypic trait at discrete life stages that affects fitness. But the true fitness value of a trait also depends on how the expression (or lack of expression) of that trait in subsequent stages affects performance, so isolating fitness metrics within single stages can give a misleading impression of overall adaptive potential [8]. For example, although size is commonly used as a fitness correlate for amphibians as they undergo metamorphosis, compensatory growth in subsequent life stages can eliminate or reverse size differences among groups, which may limit the overall impact of metamorphic size on the lifetime fitness of the individual. Here, we present a new conceptual and experimental framework that bridges the evo-devo and eco-evo viewpoints with a population genetic perspective to form a groundwork upon which we can build a more complete understanding of how stage-specific selection may

**Box 1.** Terminology that encapsulates eukaryotic life cycle diversity.

*Genetic correlations and pleiotropy:* When two or more traits appear to be inherited together (i.e. they are genetically correlated), it can indicate either that their trait values are affected by a single allele (which is known as *pleiotropy*), or by alleles at two or more loci that are physically close to one another on the chromosome and have correlations among their alleles (known as *linkage disequilibrium*). Pleiotropy and linkage disequilibrium are independent mechanisms potentially generating genetic correlations that influence the response of phenotypes to selection.

   *Life cycles:* A life cycle is a conceptual model of the changes an organism undergoes between a given developmental point (e.g. fertilization) and the same developmental point in the next generation. Life cycles differ in the number and nature of morphological and/or physiological transitions (life stages) as a consequence of variation in the extent of growth, remodelling, and/or differentiation of cell types among defined stages [2].

   *Life cycle variation:* Bell [2] outlined three types of life cycles: diplontic, haplontic and haplodiplontic (figure B1). In *diplontic* life cycles (e.g. animals), fertilization directly follows meiosis such that somatic development only occurs in the diploid stage (sea otter, urchin and crab in figure B1; [2]). But in *haplontic* life cycles (e.g. charophytes not shown in figure B1, but like diplontic life cycles), meiosis directly follows fertilization, such that somatic development only occurs in the haploid stage. In *haplodiplontic* life cycles (e.g. kelp in figure B1), meiosis and fertilization are spatiotemporally separated such that somatic development occurs in both haploid and diploid stages.

   *Life-history traits:* Life-history traits vary across individuals, populations, species and environments and describe patterns of survival, growth, maturation and fecundity that influence the demography and growth dynamics of a population [12]. Selection acts to maximize fitness across the *life cycle*, integrating the entire reproductive performance of an individual whereby the *life-history traits* are the major phenotypic components of fitness [12,13]. *Synthesis:* Confusion often arises due to the colloquial use of 'life history' as a synonym for life cycle and life-history traits (box 2). We reinforce the use of 'life history' only in the context of describing specific life-history traits, whereas a description of the history of a particular organism's progress through its life stages is referred to only as a life cycle. By focusing on the timing of meiosis and fertilization in the life cycle, accompanied by the number of stages in the cycle, we can then map life-history traits onto diverse eukaryotic life cycles.

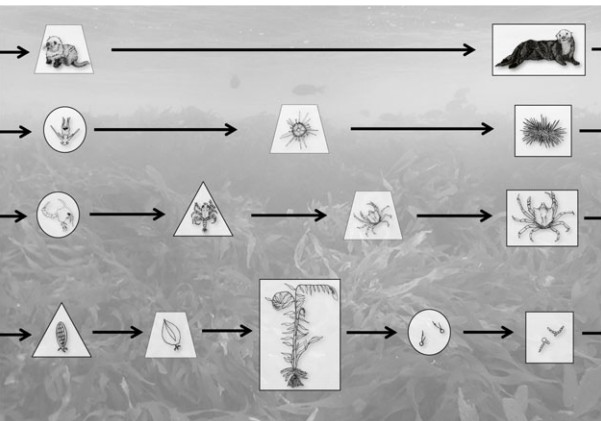

**Figure B1.** To illustrate a range of diversity found in a single, well-known ecosystem, we show (from top to bottom) the life cycles of a sea otter, an urchin, a crab and a kelp, though we note that not all life cycle stages occupy the kelp forest habitat (image background) exclusively. Different shapes refer to different life cycle stages: larval or spore (circle), additional larval or early development stage (triangle), juvenile (trapezoid) and adult (square/rectangle). Diplontic life cycles are shown here by a sea otter, urchin and kelp crab. Haplontic life cycles (not shown here but can be found in dinoflagellate lineages) will be most like the otter life cycle as there is only a free-living haploid stage. Haplodiplontic life cycles, shown by the giant kelp, have dispersive zoospores, early developmental sporophytes (triangle) and juvenile stages that lead to the adult sporophyte (shown in centre; we do not show juvenile gametophytes). Gametophytes (far right square) are microscopic. Background image is of an *Eisenia* (*Ecklonia*) *arborea* kelp forest at Santa Catalina Island (photo credit: SA Krueger-Hadfield) and line drawings were provided by Kathryn Schoenrock.

promote or constrain adaptation to environmental change. We present a genomic framework because many species do not have traits that are directly comparable from one life stage to the next (e.g. oral arm on an urchin pluteus larvae versus spine length on an adult urchin), whereas an allele is a unit that is directly comparable from one life stage to the next. Second, many species have traits relevant to climate change that are difficult to measure at the phenotypic level but easier to measure at the genomic level among life stages. For example, thermal tolerance between a kelp gametophyte versus sporophyte would be difficult to measure given little phenotypic overlap, whereas expression of heat shock proteins would be directly comparable.

# 3. Conceptual framework

Selection can act on different life stages in a single organism with a complex life cycle. The consequences may be decoupled, antagonistic, or synergistic [17]. When a fitness

**Box 2.** Literature review of articles investigating selection on species with complex life cycles.

For the literature review of previous studies investigating selection on species with complex life cycles, the methodology, inclusion criteria and a database containing all studies see the electronic supplementary materials, S3 and table S1. Briefly, we searched the following terms: 'complex life cycle' AND 'selection' AND 'life stage' in 11 databases. This search resulted in more than 900 peer-reviewed articles. We further refined studies including the search terms 'evolution' or 'adaptation' or 'embryo' or 'larva' or 'juvenile', and we limited them to papers published between 1980 and 2020. Studies ($n = 121$) included in the review used a broad collection of study organisms, mostly represented by marine invertebrates, terrestrial insects and amphibians (figure B2). Several articles applied theoretical approaches on hypothetical organisms that metamorphose from a larval stage to an adult stage. Search terms also generated a significant number of empirical articles on parasites. Parasites have complex life cycles as they depend on the exploitation and infection of one or more hosts to complete their life cycle and often alternate between sexual and asexual reproduction [14]. Platyhelminths, annelids and nematode worms seem to be especially ideal for studying the selection at different life stages (e.g. transmission rates) because their phyla include a diverse range of reproductive modes and evolutionarily independent switches [15,16]. More than half of all authors used the terms 'life-history traits' and 'life cycles' interchangeably (see box 1). Most studies in the literature search, including those based on parasites with more than two hosts, were based on only two different life stages, except for marine invertebrates, especially crustaceans, where selection across more than two life stages was considered.

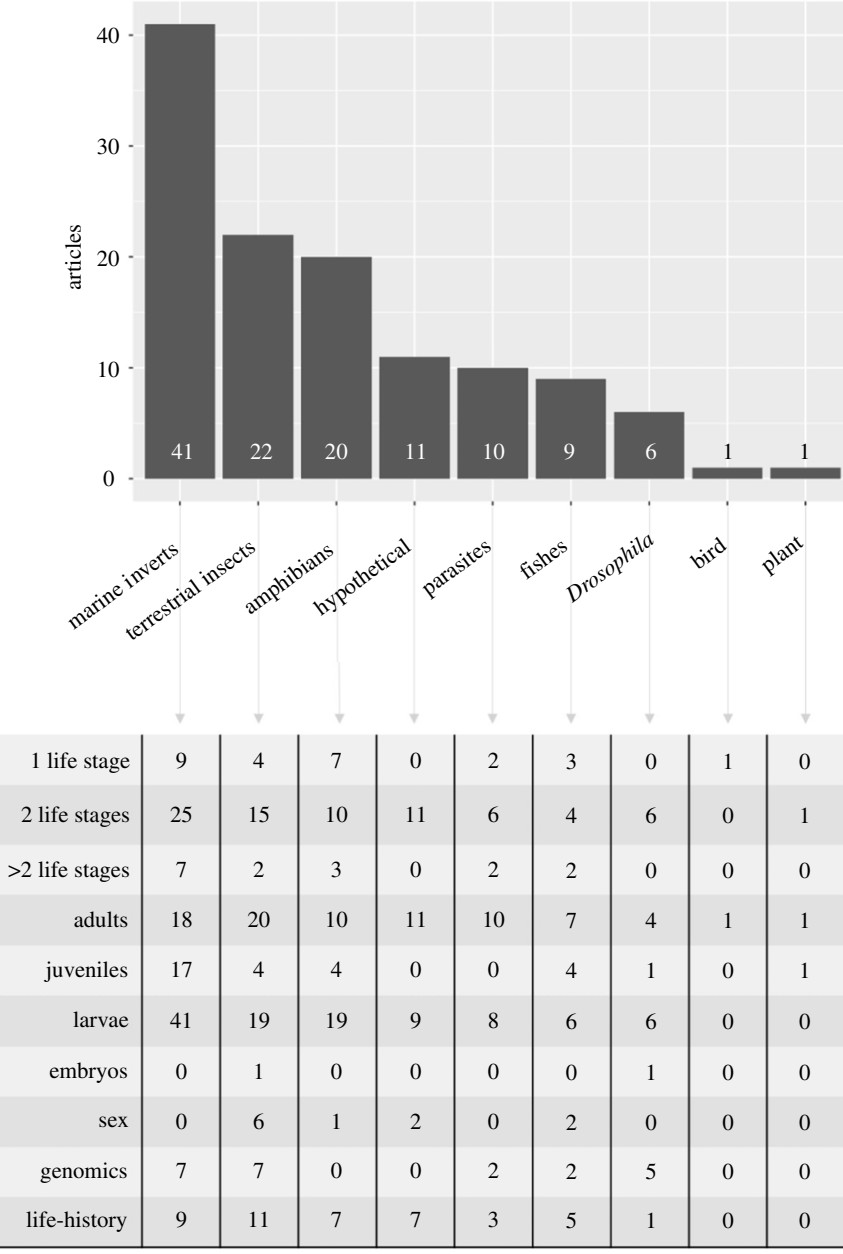

|  | marine inverts | terrestrial insects | amphibians | hypothetical | parasites | fishes | *Drosophila* | bird | plant |
|---|---|---|---|---|---|---|---|---|---|
| 1 life stage | 9 | 4 | 7 | 0 | 2 | 3 | 0 | 1 | 0 |
| 2 life stages | 25 | 15 | 10 | 11 | 6 | 4 | 6 | 0 | 1 |
| >2 life stages | 7 | 2 | 3 | 0 | 2 | 2 | 0 | 0 | 0 |
| adults | 18 | 20 | 10 | 11 | 10 | 7 | 4 | 1 | 1 |
| juveniles | 17 | 4 | 4 | 0 | 0 | 4 | 1 | 0 | 1 |
| larvae | 41 | 19 | 19 | 9 | 8 | 6 | 6 | 0 | 0 |
| embryos | 0 | 1 | 0 | 0 | 0 | 0 | 1 | 0 | 0 |
| sex | 0 | 6 | 1 | 2 | 0 | 2 | 0 | 0 | 0 |
| genomics | 7 | 7 | 0 | 0 | 2 | 2 | 5 | 0 | 0 |
| life-history | 9 | 11 | 7 | 7 | 3 | 5 | 1 | 0 | 0 |

**Figure B2.** Results from the literature review of articles investigating selection on species with complex life cycles. Histogram (top) shows the number of articles found for each group. Lower table shows the number of articles published on different life stages, the number of life stages included, whether sex was considered, genomic methods were applied, and the term 'life-history' was used.

component (allele or its resulting trait value) that affects within-stage fitness at one stage is neutral at other stages (i.e. does not affect within-stage fitness), the impact of selection is decoupled because the effects are isolated to a single life stage. We refer to this as 'ontogenetic decoupling'. We opt to use a different term from adaptive decoupling, as we describe processes that are not necessarily adaptive. By contrast, when the fitness impacts extend beyond a single life stage, genetic correlations across traits in different life stages are expected and pleiotropy is a possible cause. Pleiotropy is typically used to indicate situations when a mutation affects multiple traits that are expressed within the same life stage [38]. We expand this to situations when an allele and resulting trait value expressed at one stage also affect that trait or another trait expressed at other life stages and refer to this as 'ontogenetic pleiotropy'. In this paper, we focus on alleles and their resulting traits that affect within-stage or total fitness (i.e. fitness components). With fitness in mind, antagonistic ontogenetic pleiotropy (AntOP) occurs when a fitness component that affects within-stage fitness within one life stage also affects within-stage fitness in the opposite direction in another life stage. Conversely, synergistic ontogenetic pleiotropy (SynOP) occurs when a fitness component that affects within-stage fitness at one life stage later affects within-stage fitness in the same direction in another life stage.

For ontogenetic decoupling, the effect on total fitness and adaptive potential is determined by the direction of selection during the life stage on which selection acts. In other words, the evolutionary processes of ontogenetic decoupling should mirror those occurring in organisms with simple (single stage) life cycles (e.g. the sea otters in box 1). On the other hand, AntOP is likely to constrain evolutionary responses to a new environment. If the overall optimum phenotype shifts (as is expected in climate change scenarios), AntOP may result in maladapted phenotypes that may cause population declines. By contrast, a fitness component under SynOP is predicted to be unconstrained by ontogenetic pleiotropy, leading to positive selection and eventual fixation (or loss) of alleles affecting the fitness component in question. Thus, SynOP can a) increase the probability of adaptation to a novel environment because the beneficial effects of fitness components that improve overall fitness are reinforced across life stages (and improving total fitness), or b) reduce the probability that organisms will persist in the novel environment if unfavourable effects are shared across life stages (thus causing total fitness declines).

The distribution of ontogenetic decoupling, AntOP and SynOP across the genome and across traits is an area of research deserving increased attention and may also shift in new environments. For example, let us imagine that in the current environment, allele A increases within-stage fitness in stage 1 and increases within-stage fitness in stage 2 (SynOP). After an environmental change, allele A increases within-stage fitness in stage 1, but now decreases within-stage fitness in stage 2. In this thought experiment, the relationship changes from SynOP to AntOP following environmental change. Therefore, to understand an organism's vulnerability to environmental change, three things must be identified and characterized: (i) the distribution of AntOP, SynOP and ontogenetic decoupling across fitness components (alleles or polygenic traits) in the current environment; (ii) the way these relationships will change in

future environments and (iii) the cumulative effect of fitness components on total fitness before and after environmental change. In the service of these goals, we present a hypothesis-testing framework that is broadly applicable to genomic approaches in non-model organisms.

## 4. Hypothesis-testing using genomics

Adaptation manifests as changes in allele frequencies and trait means in response to selection over multiple generations. Evolve-and-resequence experiments, where selection is applied across multiple generations and paired with iterative genomic sequencing, are commonly used to test for selection responses across generations [39]. However, if one substitutes stages for generations, the general logic of evolve-and-resequence designs can be applied to within-generation experiments for organisms with complex life cycles in which allele frequencies and trait means are measured in response to selective pressure over multiple stages. As such, within-generation experiments can be complementary to long-term studies by efficiently measuring the degree and direction of genetic correlations across life stages, even in longer-lived species or species difficult to breed in captivity. The term 'adaptation' here refers specifically to responses to selection, though we acknowledge that experiments can say less about the multidimensional processes of adaptation in nature, and within-generation responses will not necessarily predict multigenerational adaptation. Nonetheless, allele frequencies and phenotypic trait means are metrics that can indicate a response to selection for fitness components within each stage or inform the potential for adaptive responses when total fitness (across the whole life cycle) is measured (see electronic supplementary material, table S2).

Within the context of a single-generation experiment, evidence for ontogenetic decoupling should produce datasets in which fitness components (specifically, allele frequencies or specific traits) change significantly in response to selection at one life stage, but do not change significantly in frequency in response to the same selection at a different life stage (table 1 and figure 1). AntOP should produce datasets in which fitness components change significantly in frequency in response to selection at one life stage but respond significantly in the opposite direction in a subsequent life stage. Finally, SynOP should demonstrate fitness components that change significantly in frequency in response to selection at one life stage and respond significantly in the same direction during a subsequent life stage.

## 5. Experimental design for detecting within-generation selection

Population genomics offers a straightforward method to measure the extent to which ontogenetic decoupling, AntOP, or SynOP promotes or constrains an organism's ability to evolve hypothesized adaptations to climate change in a within-generation experimental design. We suggest an experimental framework (outlined in figure 1a) that compares genome-wide allele frequency dynamics between control and treatment conditions across different stages within a life cycle. Figure 1a presents an ideal but simple scenario in which the experimental treatment does not change

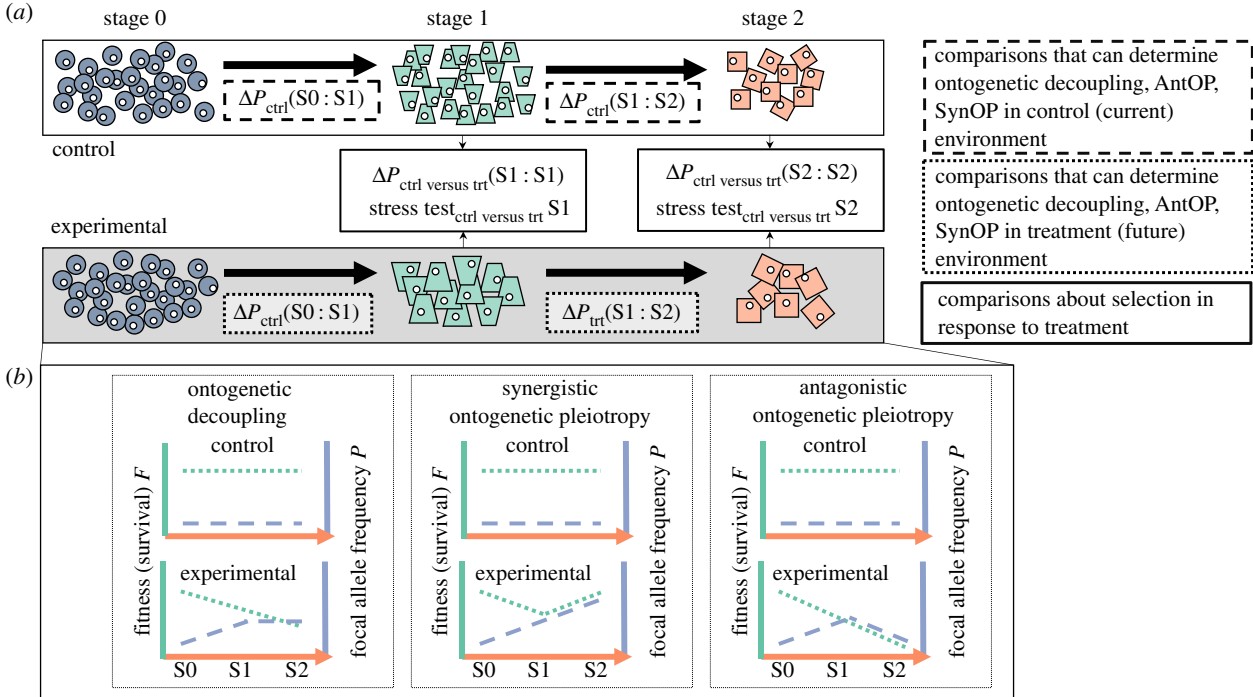

**Figure 1.** Depiction of a general framework (*a*) and expectations (*b*) when testing for changes in allele frequency in response to selection across life stages within a single generation. $\Delta P$ is the change in allele frequency between stages. Different shapes indicate different morphologies of a species as it passes through the different phases of its complex life cycle (box 1 and figure B1). In stage 1, individuals are exposed to control or experimental treatments. For the purposes of this diagram, we assume that the experimental treatment reduces survival (fitness) to the same extent in stages 1 and 2 but the focal allele increases survival in stage 1 and potentially in stage 2, depending on the hypothesis. Thus, the focal allele increases in frequency in the experimental treatment in stage 1 for all hypotheses. Both groups are reared through stage 2 in the same treatments, with fitness and allele frequencies remeasured. The control treatment is not subject to a selective pressure, so the survival (fitness) and allele frequencies do not change under any hypothesis. (Online version in colour.)

**Table 1.** Hypotheses, predictions and supporting data on outcomes of selection across life stages and impacts on subsequent stages in species with complex life cycles.

| hypothesis | general prediction | evidence in support of prediction in population genetic framework |
|---|---|---|
| *ontogenetic decoupling* | fitness components (alleles/traits) experiencing strong selection at one life stage are neutral at another life stage experiencing the same selection pressure | significant change in frequency in response to selection at one life stage where they are likely expressed, but do not change in frequency with selection at a different life stage where they may or may not be expressed |
| *synergistic ontogenetic pleiotropy (SynOP)* | fitness components (alleles/traits) experiencing strong selection at one life stage also experience strong selection in the same direction in a subsequent life stage | significant change in frequency in response to selection at one life stage, change frequency in the same direction during selection in a subsequent life stage, and are likely expressed at both life stages |
| *antagonistic ontogenetic pleiotropy (AntOP)* | fitness components (alleles/traits) experiencing strong selection at one life stage, experience strong selection in the opposite direction in a subsequent life stage | significant change in frequency in response to selection at one life stage, change frequency in the opposite direction during selection in a subsequent life stage, and are likely expressed at both life stages |

development time to a particular stage; in the case that development time is affected by the treatment, additional controls for time of sampling may need to be collected. In some taxa and stages, individual sequencing is logistically challenging (e.g. marine larvae or other microscopic stages). To accommodate diverse taxa, our experimental design assumes pooled sequencing of whole genomes (e.g. 'pool-seq': multiple individuals are pooled into a single sample for an allele

frequency estimate; electronic supplementary material, S1; [40]). However, sequencing of individuals may provide advantages in some taxa (e.g. haplotype information, linkage; see [41] and [42] for further discussion on pooled versus individual sequencing).

We use this simple design to provide a baseline of the experimental requirements needed to effectively study ontogenetic decoupling, AntOP, SynOP, and how these

relationships affect responses to climate change. However, it is worth noting that this design can easily be modified to allow for more complex designs (such as fully crossed, factorial designs). Our design specifically uses environmental treatments to look at the genetic effect of stressors, contrasting with the approach of [17], in which environmental effects and plasticity are controlled to reveal genetic effects. If environmental effects (plasticity) or gene-by-environment interactions are of interest to researchers, this experimental design can be expanded from the current two-environment design to an environmental gradient/dose-response design that improves resolution of environmental effects on ontogenetic decoupling, AntOP, SynOP across life stages.

Issues of density, developmental rates and genetic diversity are important considerations in the implementation of this design [43]. For example, when environmental stressors cause differential mortality across pools of individuals, density-dependent changes to developmental rates may be expected to confound experimental results if not accounted for. Moreover, overly strong selective regimes (e.g. high mortality) may deplete genetic variation in pleiotropic alleles and mask subsequent ontogenetic pleiotropy. We encourage the use of pilot studies to delineate how much stress can be applied so that numbers (and genetic variation) are not overly depleted. In addition, we encourage the use of species with high fecundity and high genetic diversity to reduce issues of depleting genetic variation (see study system considerations). To control for different developmental rates, we suggest collecting additional samples that account for both time and developmental stage. We acknowledge that these solutions complicate the experimental design and remain an issue for selection studies that warrants broader discussion. Nonetheless, these solutions can preserve the ability to gain insights from short-term experimental designs.

## 6. Power analysis

We used simulations to explore the experimental designs (figure 1a) that would have power to detect changes in allele frequency. We calculated power as the proportion of times the observed allele frequency change was outside the 95% quantiles of the null distribution. Because the analysis is based on permutation, the false positive rate is fixed at 5%. Methods for simulations are available in electronic supplementary materials, S1 and R Markdown script can be used to explore different experimental designs.

Published within-generation experimental designs range from approximately 50 individuals with analysis focused on a single locus [44], approximately 140 individuals sequenced genome-wide [45], or a pooled sample of 1000 individuals with unreported read depth with RNA-seq [46]. These studies typically observe allele frequency changes of less than 0.1, with significant changes in allele frequency due to selection ranging between 0.05 and 0.15 [44–46]. Based on these studies, we were primarily interested in scenarios when the power to detect an increase in allele frequency ($\Delta P = p_1 - p_0$; figure 1a) of 0.1 exceeded 80% across various levels of initial minor allele frequency ($p_0$; figure 1).

Simulations show that the power to detect an increase in allele frequency from one time point to the next is higher when the initial minor allele frequency ($p_0$) is rare (dark blue versus light blue; figure 2). Generally, power increases

at a greater rate with increased number of sampled individuals compared to increased read depth (figure 2) because there is little benefit of increasing the read depth beyond the number of chromosomes sampled from the population (figure 2c versus e). The power to detect an allele frequency increase of 0.1 exceeds 80% across all levels of initial minor allele frequency when 10 000 individuals are present in each of the initial pools and the read depth is 200× (figure 2). Thus, some empirical within-generation selection experiments [46–48] are probably drastically under-powered to detect within-generation changes in allele frequency due to selection, except for alleles that are initially rare in the population and increase in frequency during the experiment (dark blue; figure 2).

## 7. Alternative experimental genomic approaches

We based the experimental design on the collection of population genomics data, although data on gene expression, quantitative genetic measures of genetic correlations and early–late fitness trade-offs could be collected and would add complementary information. For example, the level of mRNA production at a gene can be measured across the genome at multiple developmental time points and provide a comprehensive spatial and temporal map of possible gene contributions to the phenotype. Even though we are far from understanding genotype–phenotype connections for most genes, the level of mRNA production is a phenotype subject to natural selection and can be measured with RNAseq, even in non-model organisms. Further discussion on the applicability of the hypothesis testing framework can be found in electronic supplementary material, S2.

## 8. Study system considerations

Complex life cycles are common in nature, but marine systems may be especially well suited to selection experiments across life stages. A high proportion of selection studies have used marine invertebrates during the past 40 years (figure B2). Furthermore, many benthic marine invertebrates (and many fishes) exhibit complex life cycles in which the larval stage(s) disperse, grow and develop in the pelagic zone, and have characteristics amenable to the sampling requirements set forth by the power analysis. Most marine invertebrates spawn freely into the water column, making gamete isolation and parentage manipulation easy [49]. Many of these species are important in aquaculture, developmental biology and toxicology research, resulting in detailed laboratory protocols for development and rearing (e.g. [50]). However, studies including genomic analyses in marine invertebrates are still relatively rare (box 2) and fewer still in marine macroalgae. Investigating questions about selection across life stages in changed environments using marine species has the added benefit of aiding with aquaculture methods, conservation and invasive species management.

## 9. Conclusion

Although complex life cycles are omnipresent in nature, we do not know whether having a complex life cycle will promote or constrain adaptation to environmental change.

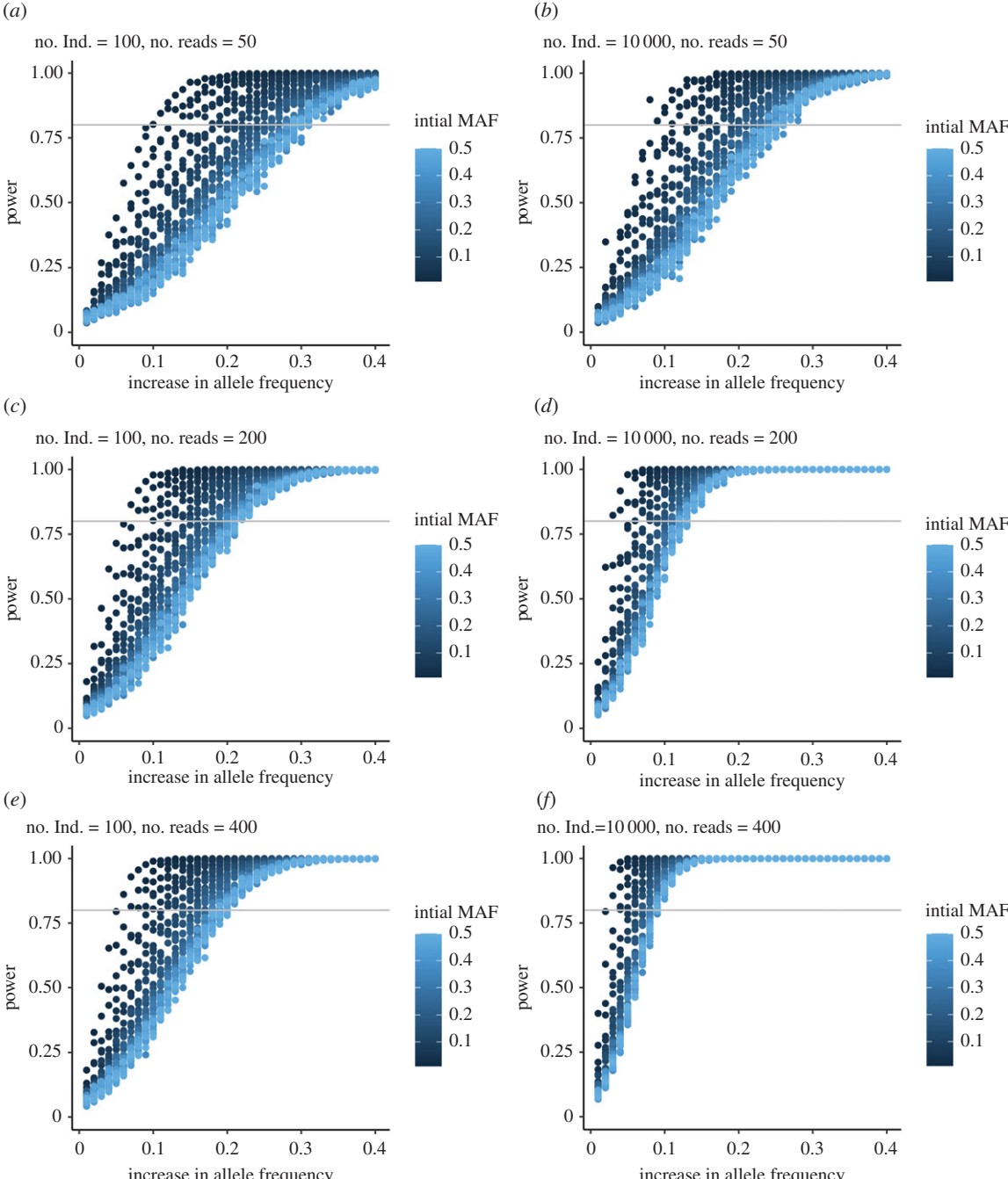

**Figure 2.** Results from the power analysis for the within-generation selection simulations. The *x*-axes represent the magnitude of allele frequency increase from before versus after selection, and the colour of the points represents the initial minor allele frequency (MAF) in the population. The horizontal grey line in each figure is at 80% power; when points are above this line the design is considered powerful enough to detect a true allele frequency increase; no. Ind refers to the number of individuals within the sample.

Here, we have outlined a framework to test hypotheses about how selection is operating across the genome in organisms with complex life cycles. We hypothesize that if the effects of selection are isolated within a particular stage (ontogenetic decoupling), having a complex life cycle should neither promote nor restrict adaptation, but that evolutionary dynamics in those instances should mirror the evolutionary dynamics of organisms with simple life cycles. However, genes that exhibit ontogenetic pleiotropy but experience selection in opposite directions across stages (AntOP) will have a constrained evolutionary response to selection. Conversely, genes that experience selection in the same direction across stages (SynOP) will be less constrained in their evolutionary response to selection. We provide a framework to tease these patterns apart but suggest that despite their

power, genomic approaches remain a precursor for testing trait evolution in nature.

Data accessibility. The data are provided in electronic supplementary material [51].

Authors' contributions. M.A.A.: conceptualization, project administration, supervision, visualization, writing–original draft, writing–review & editing; L.G.E.W.: conceptualization, data curation, formal analysis, investigation, supervision, visualization, writing–original draft, writing–review & editing; S.A.K.-H.: conceptualization, visualization, writing–original draft, writing–review & editing; S.M.B.: conceptualization, visualization, writing–original draft, writing–review & editing; M.W.H.: conceptualization, writing–original draft, writing–review & editing; M.P.H.: conceptualization, writing–original draft, writing–review & editing; H.K.K.: conceptualization, writing–original draft, writing–review & editing; M.A.S.: conceptualization, writing–original draft, writing–review & editing; K.E.L.:

conceptualization, formal analysis, funding acquisition, investigation, methodology, visualization, writing–original draft, writing–review & editing; A.M.R.: conceptualization, project administration, supervision, writing–original draft, writing–review & editing. All authors gave final approval for publication and agreed to be held accountable for the work performed therein.

Competing interests. The authors declare no conflicts of interest.

Funding. Financial support for the workshop and for M.A.A. was provided by NSF-OCE 1764316 to K.E.L. L.G.E.W. was supported by funding from the European Union's Framework Programme for Research and Innovation Horizon 2020 (2014-2020) under the Marie Sklodowska-Curie grant agreement no. 101025649.

Acknowledgements. We thank Will Ryan for helpful discussion and Kathryn Schoenrock for drawings in figure B1.

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
