## [Peer Review File · Proceedings of the Royal Society B: Biological Sciences]

Review History

RSPB-2021-0756.R0 (Original submission)

Review form: Reviewer 1

Recommendation

Accept with minor revision (please list in comments)

Scientific importance: Is the manuscript an original and important contribution to its field?

Excellent

General interest: Is the paper of sufficient general interest?

Excellent

Quality of the paper: Is the overall quality of the paper suitable?

Excellent

Is the length of the paper justified?

Yes

Should the paper be seen by a specialist statistical reviewer?

No

Do you have any concerns about statistical analyses in this paper? If so, please specify them explicitly in your report.

No

It is a condition of publication that authors make their supporting data, code and materials available - either as supplementary material or hosted in an external repository. Please rate, if applicable, the supporting data on the following criteria.

Is it accessible?

Yes

Is it clear?

Yes

Is it adequate?

Yes

Do you have any ethical concerns with this paper?

No

Comments to the Author

How selection may be linked (or not) across life stages is a fascinating topic that has largely been neglected in the evolutionary literature. This manuscript ambitiously aims to bring attention to the topic, clarify thinking and terminology across disciplines, and suggest an experimental framework for future empirical research. Overall the paper succeeds in this goal. The content is strong and the writing is notably smooth and persuasive. This is an excellent and thought provoking piece of work that I am sure to refer back to in the future and recommend to my students.

The only conceptual/analytical concerns I have are that: 1) an overly strong selective regime could deplete genetic variation such that the subsequent ontogenetic pleiotropy is masked, 2) simulations can guide experiments with respect to power, but consideration of false positive rates would be equally useful to empirical biologists. If the authors consider these points to have merit, then incorporating those ideas into the paper would be helpful.

My main comments are aimed at clarifications and presentation. This is one of the few papers that reads as overly short and in various places concrete biological examples are neglected. I wonder if this brevity is due to journal word restrictions - if that is the case, I would encourage the journal editors to allow some latitude because the paper will be stronger with a slight increase in length. Specifically, the paper's topic would be more compelling if the ubiquity of discrete life stages across the tree of life was made obvious within the first few paragraphs. Similarly, the paragraphs on the developmental constraint hypothesis, eco-evolutionary perspective, and population genetics presents as quite vague: adding some concrete biological examples would underscore the points being made and animate the paper. For instance: "...there is experimental support for this Ho for taxa x, y, and z and no support for r, s, t but this hypothesis has not been comprehensively evaluated for taxa with life cycles that include...."

The supplemental should supply more information on the literature survey results - full species names should be provided ("marine invertebrate" is extremely very vague). Reporting the DOI is fine, but a full citation should be provided too. And another remark aimed to the journal: the journal should figure out a way to provide citation "credits" to original data producers.

In summary: I really like this paper and my comments are suggestions aimed to strengthen its relevancy to empirical biologists. The journal could consider relaxing strict word limits if required and should credit all the papers that have informed this manuscript.

Minor comments:

100 consider explicitly defining pleiotropy at first mention because different readers may have subtly different mental models for this term

100-101 missing word?

117, 130 Are pleiotropy and adaptive decoupling different points along a continuum (extent to which a loci has multiple phenotypic outcomes) or are they fundamentally different? Clarify.

136 I agree with this statement but it would have been nice early on to establish the prevalence of this phenomenon with biological examples.

150 -151 This clarity is what I was looking for in the previous section!

153 perfect clear definition

189 follow up with a definition/ example of the evolve and re-sequence experimental design for readers that might not be familiar with that construct

192 follow up with an example of what an across life stage experiment might look like
Simulations - what is the rate of false positives?

Fig B1 - Although I like this figure conceptually, some modifications will highlight the nuance a bit more. For the marine example, depicting the duration and habitat used for each stage will increase accuracy and (with correct habitat identified) underscore the sometimes profound differences in selective challenges faced by different stages. The kelp background, while attractive, implies that all development occurs there, which is not true. Highly disparate stages are probably more likely to incur trade-offs. Also, in keeping with the spirit of the paper's intent to unify thinking across environments, a terrestrial example would be a welcome complement to the kelp forest example.

Fig 1 - I do not understand why you expect fitness in the experimental to decrease at Stage 2 under ontogenetic decoupling; shouldn't the fitness of experimental and control look the same here (slope = 0)? Similarly under synOP I would expect fitness in experimental to be higher (conditioned on the S1->S2 transition). The figure legend restates the expectation shown in the figure but does not explain why.

Is there a concern that if overly strong selection is induced at S1 that variation is depleted and therefore any OP response could be masked?

Overall I quite like this figure and appreciate that it includes both a concept diagram and pseudo data expectations.

Cynthia Riginos

Review form: Reviewer 2 (André de Roos)

Recommendation

Reject – article is scientifically unsound

Scientific importance: Is the manuscript an original and important contribution to its field?

Poor

General interest: Is the paper of sufficient general interest?

Marginal

Quality of the paper: Is the overall quality of the paper suitable?

Poor

Is the length of the paper justified?

Yes

Should the paper be seen by a specialist statistical reviewer?

No

Do you have any concerns about statistical analyses in this paper? If so, please specify them explicitly in your report.

No

It is a condition of publication that authors make their supporting data, code and materials available - either as supplementary material or hosted in an external repository. Please rate, if applicable, the supporting data on the following criteria.

Is it accessible?

Yes

Is it clear?

Yes

Is it adequate?

Yes

Do you have any ethical concerns with this paper?

No

Comments to the Author

See the attached PDF file. (See Appendix A)

Review form: Reviewer 3

Recommendation

Major revision is needed (please make suggestions in comments)

Scientific importance: Is the manuscript an original and important contribution to its field?

Good

General interest: Is the paper of sufficient general interest?

Good

Quality of the paper: Is the overall quality of the paper suitable?

Acceptable

Is the length of the paper justified?

Yes

Should the paper be seen by a specialist statistical reviewer?

No

Do you have any concerns about statistical analyses in this paper? If so, please specify them explicitly in your report.

No

It is a condition of publication that authors make their supporting data, code and materials available - either as supplementary material or hosted in an external repository. Please rate, if applicable, the supporting data on the following criteria.

Is it accessible?

Yes

Is it clear?

Yes

Is it adequate?

Yes

Do you have any ethical concerns with this paper?

No

Comments to the Author

In this review, Albrecker et al. synthesize the importance of integrating the effects of selection across life cycles when considering adaptation to environmental change. They then propose an experimental framework for using genomics to test how the fitness effects of alleles vary across stages and conduct simulations to provide guidelines for appropriate sample sizes and sequencing coverage.

Considering selection across the entire life cycle is an extremely important and often overlooked component of adaptation. Additionally, I think the idea of applying evolve and resequence thinking to within-generation selection is quite clever. I expect that this paper could be used a framework for many studies looking to understand the selection across complex life cycles.

While I think the concept is extremely interesting and important, I did feel that there were large holes in the 'review' component of the paper and the text could be generally streamlined. I understand the authors may be space limited, but I think that the background and box sections could be used to provide researchers more context for what we already know about trait correlations across metamorphic boundaries. Although the following are purely my opinion, here are some specific ideas for what I felt could be added/removed:

- 1) For me, the presentation of the separate fields (evo-devo, eco-evo, population genetics) that are meant to be synthesized did not add much to the paper. In fact, I felt some of the most important contributions of each were overlooked. For example, the field of evo-devo has elucidated the function of many important genes. Also, the paragraph on population genetics reads more like a paragraph on quantitative genetics, highlighting correlations in trait expression.
- 2) While I appreciate that the authors are trying to highlight the diversity of life cycles and clarify some terminology, I felt that Box 1 was largely tangential to the main point of the paper.

3) In place of some of the above content, I would have liked more of a true review of correlations in traits across metamorphosis. This is super important for establishing the need to understand the variation selective affects. I've listed a few examples at the end of this review.

4) Similar to above, the authors are not the first to consider pleiotropy across developmental stages (see for example Donohue 2014) and I would like to see more previous work discussed. Highlighting what we know about this field is again important to the main argument of the paper.

For the experimental design, I'm concerned about the lack of a proper control for developmental plasticity. You can imagine, for example, that epigenetic (or other) effects before stage 1 might alter the fitness consequences of particular alleles. Although this may be 'realistic' if organisms are in a constant environment across their life cycles, it is less so in fluctuating environments. To get a more 'true' measure of the fitness affects of each allele at each stage, it seems like you would need a fully crossed design.

Decision letter (RSPB-2021-0756.R0)

Dear Ms Albecker:

I am writing to inform you that your manuscript RSPB-2021-0756 entitled "Does a complex life cycle affect adaptation to environmental change? Genome-informed approaches for characterizing selection across life cycle stages" has, in its current form, been rejected for publication in Proceedings B.

This action has been taken on the advice of referees, who have recommended that substantial revisions are necessary. With this in mind we would be happy to consider a resubmission, provided the comments of the referees are fully addressed. However please note that this is not a provisional acceptance.

Sincerely,
Dr Sasha Dall
mailto: proceedingsb@royalsociety.org

Associate Editor
Board Member: 1
Comments to Author:
Dear Dr. Albecker,

I have now received three reviews of your manuscript “Does a complex life cycle affect adaptation to environmental change? Genome-informed approaches for characterizing selection across life cycle stages”. As you will see, the reviewers were quite divergent in their opinion of the manuscript with one recommending Rejection, one recommending Accept with Minor Revisions, and the other recommending Major Revision.

Based on my own reading of the manuscript, I found the various reviewer comments sufficiently helpful and substantive to inform a major revision of the MS. Yet I am also concerned about the potentially high degree of overlap between your paper with Collet and Fellous (2019) that also appeared in Proc B. Given this potential overlap, I am recommending Reject and Allow Resubmission so that you have the additional and sufficient time to identify and highlight the distinct contributions made by your paper.

My sense is that the success of a resubmission would be contingent upon your ability to address this concern as well as your ability to effectively address the other reviewer comments. Please note that I do not expect you to add unnecessary length to your MS; some of the comments by the reviewers could be viewed as being beyond the scope of your paper and therefore do not warrant substantial treatment. Nevertheless, you should be clear as to why you chose to include or exclude manuscript content that addresses these points.

In closing, I want to emphasize that I am enthusiastic about receiving a revised version. I hope that the additional time afforded by this decision will allow you to revise your manuscript so that it can be included in our Special Feature.

Cheers,
Geoff

Reviewer #1

While this reviewer was quite supportive and offered several comments for you to consider. I note that this reviewer agreed with R#3 that your coverage of evo-devo, eco-evo and population genetics was vague and could benefit from some concrete examples.

Reviewer #2

This reviewer was obviously was the most critical and raised a number of important points that should be addressed if the paper is to be successful. The reviewer’s comments are largely self-evident but I did want to highlight some key comments. First, the reviewer felt that this paper does not provide significant advances beyond those offered by Collet and Fellous (2019) in Proc B, which addresses themes similar to those covered in your paper. Second, the reviewer felt that there were potential internal inconsistencies between your adoption of “the entire life cycle as the central unit” and the three hypotheses that emphasize “fitness within a stage”; this point is reinforced further with comments about the limitations with the evolve and resequence approach within a single generation. The third concern is the lack of attention to environmental effects. Finally, the reviewer expressed concern about the ignoring of demographic coupling which is central to complex life cycles and the potentially significant effects of metamorphosis on individual fitness.

Reviewer #3

This reviewer was more enthusiastic about the conceptual value of the paper but also felt that there were “large holes” in the review. First, the reviewer felt that your synthesis of evo-devo, eco-evo and population genetics did not add much to the paper. Second, they felt that the paper could be improved by a deeper review of trait correlations across metamorphosis. Finally, the reviewer felt that you could do a better job of discussing previous work that considers pleiotropy across developmental stages.

Reviewer(s)' Comments to Author:

Referee: 1

Comments to the Author(s)

How selection may be linked (or not) across life stages is a fascinating topic that has largely been neglected in the evolutionary literature. This manuscript ambitiously aims to bring attention to the topic, clarify thinking and terminology across disciplines, and suggest an experimental framework for future empirical research. Overall the paper succeeds in this goal. The content is strong and the writing is notably smooth and persuasive. This is an excellent and thought provoking piece of work that I am sure to refer back to in the future and recommend to my students.

The only conceptual/analytical concerns I have are that: 1) an overly strong selective regime could deplete genetic variation such that the subsequent ontogenetic pleiotropy is masked, 2) simulations can guide experiments with respect to power, but consideration of false positive rates would be equally useful to empirical biologists. If the authors consider these points to have merit, then incorporating those ideas into the paper would be helpful.

My main comments are aimed at clarifications and presentation. This is one of the few papers that reads as overly short and in various places concrete biological examples are neglected. I wonder if this brevity is due to journal word restrictions – if that is the case, I would encourage the journal editors to allow some latitude because the paper will be stronger with a slight increase in length. Specifically, the paper's topic would be more compelling if the ubiquity of discrete life stages across the tree of life was made obvious within the first few paragraphs. Similarly, the paragraphs on the developmental constraint hypothesis, eco-evolutionary perspective, and population genetics presents as quite vague: adding some concrete biological examples would underscore the points being made and animate the paper. For instance: “...there is experimental support for this Ho for taxa x, y, and z and no support for r, s, t but this hypothesis has not been comprehensively evaluated for taxa with life cycles that include....”

The supplemental should supply more information on the literature survey results – full species names should be provided (“marine invertebrate” is extremely very vague). Reporting the DOI is fine, but a full citation should be provided too. And another remark aimed to the journal: the journal should figure out a way to provide citation “credits” to original data producers.

In summary: I really like this paper and my comments are suggestions aimed to strengthen its relevancy to empirical biologists. The journal could consider relaxing strict word limits if required and should credit all the papers that have informed this manuscript.

Minor comments:

100 consider explicitly defining pleiotropy at first mention because different readers may have subtly different mental models for this term

100-101 missing word?

117, 130 Are pleiotropy and adaptive decoupling different points along a continuum (extent to which a loci has multiple phenotypic outcomes) or are they fundamentally different? Clarify.

136 I agree with this statement but it would have been nice early on to establish the prevalence of this phenomenon with biological examples.

150 -151 This clarity is what I was looking for in the previous section!

153 perfect clear definition

189 follow up with a definition/ example of the evolve and re-sequence experimental design for readers that might not be familiar with that construct

192 follow up with an example of what an across life stage experiment might look like

Simulations - what is the rate of false positives?

Fig B1 - Although I like this figure conceptually, some modifications will highlight the nuance a bit more. For the marine example, depicting the duration and habitat used for each stage will increase accuracy and (with correct habitat identified) underscore the sometimes profound differences in selective challenges faced by different stages. The kelp background, while attractive, implies that all development occurs there, which is not true. Highly disparate stages are probably more likely to incur trade-offs. Also, in keeping with the spirit of the paper's intent to unify thinking across environments, a terrestrial example would be a welcome complement to the kelp forest example.

Fig 1 - I do not understand why you expect fitness in the experimental to decrease at Stage 2 under ontogenetic decoupling; shouldn't the fitness of experimental and control look the same here (slope = 0)? Similarly under synOP I would expect fitness in experimental to be higher (conditioned on the S1->S2 transition). The figure legend restates the expectation shown in the figure but does not explain why.

Is there a concern that if overly strong selection is induced at S1 that variation is depleted and therefore any OP response could be masked?

Overall I quite like this figure and appreciate that it includes both a concept diagram and pseudo data expectations.

Cynthia Riginos

Referee: 2

Comments to the Author(s)

See the attached PDF file

Referee: 3

Comments to the Author(s)

In this review, Albrecker et al. synthesize the importance of integrating the effects of selection across life cycles when considering adaptation to environmental change. They then propose an experimental framework for using genomics to test how the fitness effects of alleles vary across stages and conduct simulations to provide guidelines for appropriate sample sizes and sequencing coverage.

Considering selection across the entire life cycle is an extremely important and often overlooked component of adaptation. Additionally, I think the idea of applying evolve and resequence thinking to within-generation selection is quite clever. I expect that this paper could be used a framework for many studies looking to understand the selection across complex life cycles.

While I think the concept is extremely interesting and important, I did feel that there were large holes in the 'review' component of the paper and the text could be generally streamlined. I

understand the authors may be space limited, but I think that the background and box sections could be used to provide researchers more context for what we already know about trait correlations across metamorphic boundaries. Although the following are purely my opinion, here are some specific ideas for what I felt could be added/removed:

- 1) For me, the presentation of the separate fields (evo-devo, eco-evo, population genetics) that are meant to be synthesized did not add much to the paper. In fact, I felt some of the most important contributions of each were overlooked. For example, the field of evo-devo has elucidated the function of many important genes. Also, the paragraph on population genetics reads more like a paragraph on quantitative genetics, highlighting correlations in trait expression.
- 2) While I appreciate that the authors are trying to highlight the diversity of life cycles and clarify some terminology, I felt that Box 1 was largely tangential to the main point of the paper.
- 3) In place of some of the above content, I would have liked more of a true review of correlations in traits across metamorphosis. This is super important for establishing the need to understand the variation selective affects. I've listed a few examples at the end of this review.
- 4) Similar to above, the authors are not the first to consider pleiotropy across developmental stages (see for example Donohue 2014) and I would like to see more previous work discussed. Highlighting what we know about this field is again important to the main argument of the paper.

For the experimental design, I'm concerned about the lack of a proper control for developmental plasticity. You can imagine, for example, that epigenetic (or other) effects before stage 1 might alter the fitness consequences of particular alleles. Although this may be 'realistic' if organisms are in a constant environment across their life cycles, it is less so in fluctuating environments. To get a more 'true' measure of the fitness affects of each allele at each stage, it seems like you would need a fully crossed design.

Author's Response to Decision Letter for (RSPB-2021-0756.R0)

See Appendix B.

RSPB-2021-2122.R0

Review form: Reviewer 1

Recommendation

Accept with minor revision (please list in comments)

Scientific importance: Is the manuscript an original and important contribution to its field?

Excellent

General interest: Is the paper of sufficient general interest?

Good

Quality of the paper: Is the overall quality of the paper suitable?

Excellent

Is the length of the paper justified?

Yes

Should the paper be seen by a specialist statistical reviewer?

No

Do you have any concerns about statistical analyses in this paper? If so, please specify them explicitly in your report.

No

It is a condition of publication that authors make their supporting data, code and materials available - either as supplementary material or hosted in an external repository. Please rate, if applicable, the supporting data on the following criteria.

Is it accessible?

Yes

Is it clear?

Yes

Is it adequate?

Yes

Do you have any ethical concerns with this paper?

No

Comments to the Author

The revised version of the manuscript has been streamlined in focus and has referred most biological examples to a recent review paper. This solution works fairly well and is helpful for maintaining a shortish format. Since the focus of the present manuscript now shifts more heavily towards “evolve and resequence” experiments, a bit of review (what sort of organisms, what sorts of experimental designs) would be welcome if there is room and appetite.

Any substantive points that I had previously raised have been appropriately addressed. A few minor points follow, aimed at improving clarity and relevancy for future readers.

Lines 130-134 I appreciate that you have slimmed down the review aspects of this manuscript. Nonetheless, in this paragraph a concrete example of a noteworthy study would really help make the abstract ideas more tangible. Just a single sentence that highlights one example where discrete life stages were examined would give a sense how experiments are done and what can be found.

139-140 Very helpful examples!

T1: Give full names to SynOP and AntOP within T1 for readers who are skimming.

Fig 1 B - why are the Y axes have coloured and hashed lines with arrowheads? This is really confusing. Also, I think your labels for the Y axes are not quite accurate. Green/fitness appears to be relative survival from previous life stage - F(S2) makes no sense for a measurement taken at S1. Similarly, blue/allele frequency is not P(S2) but P(at each life stage).

Please proofread and correct your literature cites.

Review form: Reviewer 3 (André de Roos)

Recommendation

Reject – article is scientifically unsound

Scientific importance: Is the manuscript an original and important contribution to its field?

Excellent

General interest: Is the paper of sufficient general interest?

Good

Quality of the paper: Is the overall quality of the paper suitable?

Good

Is the length of the paper justified?

Yes

Should the paper be seen by a specialist statistical reviewer?

No

Do you have any concerns about statistical analyses in this paper? If so, please specify them explicitly in your report.

No

It is a condition of publication that authors make their supporting data, code and materials available - either as supplementary material or hosted in an external repository. Please rate, if applicable, the supporting data on the following criteria.

Is it accessible?

Yes

Is it clear?

Yes

Is it adequate?

Yes

Do you have any ethical concerns with this paper?

No

Comments to the Author

The authors have done a terrific job responding to reviews and I believe the paper is much improved. I have no further concerns.

Decision letter (RSPB-2021-2122.R0)

29-Oct-2021

Dear Ms Albecker

I am pleased to inform you that your manuscript RSPB-2021-2122 entitled "Does a complex life cycle affect adaptation to environmental change? Genome-informed approaches for

characterizing selection across life cycle stages" has been accepted for publication in Proceedings B.

The referee(s) have recommended publication, but also suggest some minor revisions to your manuscript. Therefore, I invite you to respond to the referee(s)' comments and revise your manuscript. Because the schedule for publication is very tight, it is a condition of publication that you submit the revised version of your manuscript within 7 days. If you do not think you will be able to meet this date please let us know.

In order to ensure effective and robust dissemination and appropriate credit to authors the dataset(s) used should be fully cited. To ensure archived data are available to readers, authors should include a 'data accessibility' section immediately after the acknowledgements section.

This should list the database and accession number for all data from the article that has been made publicly available, for instance:

Sincerely,
Dr Sasha Dall
mailto:proceedingsb@royalsociety.org

Associate Editor
Board Member
Comments to Author:
Dear Dr. Albecker,

I am pleased to report that I have received two reviews of your revised manuscript, "Does a complex life cycle affect adaptation to environmental change? Genome-informed approaches for characterizing selection across life cycle stages". Both reviewers agreed that you did a nice job addressing comments on your first submission and recommend acceptance of your paper for publication. I concur. Please note that one reviewer has some minor suggestions that you may want to consider as you prepare your final manuscript.

Congratulations on developing a nice paper and I look forward to seeing it "in print"!

Cheers,
Geoff

Reviewer(s)' Comments to Author:
Referee: 1
Comments to the Author(s).

The revised version of the manuscript has been streamlined in focus and has referred most biological examples to a recent review paper. This solution works fairly well and is helpful for maintaining a shortish format. Since the focus of the present manuscript now shifts more heavily towards "evolve and resequence" experiments, a bit of review (what sort of organisms, what sorts of experimental designs) would be welcome if there is room and appetite.

Any substantive points that I had previously raised have been appropriately addressed. A few minor points follow, aimed at improving clarity and relevancy for future readers.

Lines 130-134 I appreciate that you have slimmed down the review aspects of this manuscript. Nonetheless, in this paragraph a concrete example of a noteworthy study would really help make the abstract ideas more tangible. Just a single sentence that highlights one example where discrete life stages were examined would give a sense how experiments are done and what can be found.

139-140 Very helpful examples!

T1: Give full names to SynOP and AntOP within T1 for readers who are skimming.

Fig 1 B - why are the Y axes have coloured and hashed lines with arrowheads? This is really confusing. Also, I think your labels for the Y axes are not quite accurate. Green/fitness appears to be relative survival from previous life stage - F(S2) makes no sense for a measurement taken at S1. Similarly, blue/allele frequency is not P(S2) but P(at each life stage).

Please proofread and correct your literature cites.

Referee: 3

Comments to the Author(s).

The authors have done a terrific job responding to reviews and I believe the paper is much improved. I have no further concerns.

Author's Response to Decision Letter for (RSPB-2021-2122.R0)

See Appendix C.

Decision letter (RSPB-2021-2122.R1)

08-Nov-2021

Dear Ms Albecker

I am pleased to inform you that your manuscript entitled "Does a complex life cycle affect adaptation to environmental change? Genome-informed approaches for characterizing selection across life cycle stages" has been accepted for publication in Proceedings B.

Data Accessibility section

Open Access

Paper charges

Sincerely,

Proceedings B

Appendix A

Manuscript number: RSPB-2021-0756
Title: Does a complex life cycle affect adaptation to environmental change?
Genome-informed approaches for characterizing selection across life cycle stages
Author(s): Albecker et al.

In this paper the authors address the issue that current understanding about the ecological and evolutionary constraints imposed by life cycle complexity is limited, despite that complex life cycles abound in nature. This is an important issue and a topic that definitely deserves more study. The authors discuss genomic approaches to study selection across the multiple stages that make up a complex life cycle and propose a framework for hypothesis testing. Despite that the topic is relevant, important and timely I cannot recommend publication of this manuscript for the following reasons:

- Most importantly, in my opinion there is significant overlap with a recent article in the Proceedings of the Royal Society on exactly the same topic (Collet J, Fellous S. 2019 Do traits separated by metamorphosis evolve independently? Concepts and methods. Proc. R. Soc. B 286: 20190445. <http://dx.doi.org/10.1098/rspb.2019.0445>). The authors might have missed this paper as they do not refer to it, but I do not see how the current manuscript advances on this earlier publication.
- One important issue that I have with the current manuscript is that the authors use “alleles” and “traits” interchangeable. This lack of a clear distinction between alleles and traits results in confusing statements. Consider as one specific example line 171-174: *“Ontogenetic decoupling, AntOP, and SynOP are likely operating on different alleles and/or traits within the same genome. The distribution of the three categories across the genome is an area of research deserving increased attention, as well as how the nature and distribution of the three categories changes as the environment changes”*. The *“distribution of traits within the same genome”* only makes sense if alleles are equated to traits.
- In contrast, in their review Collet & Fellous use traits throughout their presentation, while allowing for genetic correlation between traits in different life history stages due to pleiotropy. I consider this latter conceptualization a much more logical approach. It implies, however, that genomic approaches are much less useful for studying natural selection in the context of complex life cycles, in contrast to what is argued in this manuscript.
- Another major problem is that the authors talk about fitness but never clearly define it. The authors start off by adopting the entire life cycle as the central unit, which makes “fitness” something that can only be assessed over the entire life cycle (as it should be). But in their conceptual framework subsequently discuss make statements like: *“a single allele or trait that affects fitness at one life stage also affects fitness at another life stage”*. In fact, the 3 “hypotheses” as the authors call them of ontogenetic decoupling, antagonistic ontogenetic pleiotropy, and synergistic ontogenetic pleiotropy rely on this “fitness-within-a-stage” concept and hence contrasts with the initial position that the entire life cycle is the central unit.
- In a similar vein, the evolve and resequence approach applied within a single generation that the authors propagate as a method of study strikes me at odds with the tenet that fitness is a characteristic of the entire life cycle. This approach only makes sense if the analysis is restricted to individual survival, as the authors do in Figure 1.
- In contrast to Collet & Fellous who consider gene x environment interactions, the authors pay almost no attention to an individual’s environment. Their focus is almost exclusively on genes, alleles and genomic approaches. I struggle with that lack of consideration for the environment.
- As a consequence of the previous point, the idea of “ontogenetic decoupling” in my opinion only makes sense in a context where there is absolutely no interaction between individuals. Reduced survival in one particular stage, for example, translates into fewer survivors in the

next stages, which means lower numbers of competitors and/or fewer partners for mating. In other words, in a population context different life stages are always coupled demographically, which only has no bearing on fitness in the rather extreme situation in which individuals do not interact with their conspecifics. This demographic coupling inherent in complex life cycle is ignored by the authors.

- Lastly, the authors do not really distinguish between life cycles with or without metamorphosis, as illustrated by their Figure B1 which contains both giant kelp, sea urchins and sea otters. In many species the complex life cycle also involves metamorphosis, which is a very costly, complete overhaul of an individual's body plan. This component of the fitness of an individual life history is totally ignored in this manuscript.

Signed,
André de Roos

Appendix B

Dear Dr. Trussell and reviewers:

We are grateful for the thorough and thoughtful reviews on our paper, “Does a complex life cycle affect adaptation to environmental change? Genome-informed approaches for characterizing selection across life cycle stages”. We have engaged each reviewer comment and performed substantial revisions to our paper, which we believe address reviewer comments and strengthen the manuscript’s message and clarity. Specific point-by-point responses to reviewer comments are in blue below. Thank you for re-considering our manuscript for publication in *Proceedings B. Special Issue on Evolution in Changing Seas*.

Sincerely,

Dr. Molly Albecker and Dr. Laetitia Wilkins on behalf of the authors

Comments to Author:

Dear Dr. Albecker,

I have now received three reviews of your manuscript “Does a complex life cycle affect adaptation to environmental change? Genome-informed approaches for characterizing selection across life cycle stages”. As you will see, the reviewers were quite divergent in their opinion of the manuscript with one recommending Rejection, one recommending Accept with Minor Revisions, and the other recommending Major Revision.

Based on my own reading of the manuscript, I found the various reviewer comments sufficiently helpful and substantive to inform a major revision of the MS. Yet I am also concerned about the potentially high degree of overlap between your paper with Collet and Fellous (2019) that also appeared in Proc B. Given this potential overlap, I am recommending Reject and Allow Resubmission so that you have the additional and sufficient time to identify and highlight the distinct contributions made by your paper.

My sense is that the success of a resubmission would be contingent upon your ability to address this concern as well as your ability to effectively address the other reviewer comments. Please note that I do not expect you to add unnecessary length to your MS; some of the comments by the reviewers could be viewed as being beyond the scope of your paper and therefore do not warrant substantial treatment. Nevertheless, you should be clear as to why you chose to include or exclude manuscript content that addresses these points.

In closing, I want to emphasize that I am enthusiastic about receiving a revised version. I hope that the additional time afforded by this decision will allow you to revise your manuscript so that it can be included in our Special Feature.

Cheers,

Geoff

Reviewer #1

While this reviewer was quite supportive and offered several comments for you to consider. I note that this reviewer agreed with R#3 that your coverage of evo-devo, eco-evo and population genetics was vague and could benefit from some concrete examples.

Reviewer #2

This reviewer was obviously the most critical and raised a number of important points that should be addressed if the paper is to be successful. The reviewer's comments are largely self-evident but I did want to highlight some key comments. First, the reviewer felt that this paper does not provide significant advances beyond those offered by Collet and Fellous (2019) in Proc B, which addresses themes similar to those covered in your paper. Second, the reviewer felt that there were potential internal inconsistencies between your adoption of "the entire life cycle as the central unit" and the three hypotheses that emphasize "fitness within a stage"; this point is reinforced further with comments about the limitations with the evolve and resequence approach within a single generation. The third concern is the lack of attention to environmental effects. Finally, the reviewer expressed concern about the ignoring of demographic coupling which is central to complex life cycles and the potentially significant effects of metamorphosis on individual fitness.

Reviewer #3

This reviewer was more enthusiastic about the conceptual value of the paper but also felt that there were "large holes" in the review. First, the reviewer felt that your synthesis of evo-devo, eco-evo and population genetics did not add much to the paper. Second, they felt that the paper could be improved by a deeper review of trait correlations across metamorphosis. Finally, the reviewer felt that you could do a better job of discussing previous work that considers pleiotropy across developmental stages.

Reviewer(s)' Comments to Author:

Referee: 1

Comments to the Author(s)

How selection may be linked (or not) across life stages is a fascinating topic that has largely been neglected in the evolutionary literature. This manuscript ambitiously aims to bring attention to the topic, clarify thinking and terminology across disciplines, and suggest an experimental framework for future empirical research. Overall the paper succeeds in this goal. The content is strong and the writing is notably smooth and persuasive. This is an excellent and thought provoking piece of work that I am sure to refer back to in the future and recommend to my students.

Thank you!

The only conceptual/analytical concerns I have are that: 1) an overly strong selective regime could deplete genetic variation such that the subsequent ontogenetic pleiotropy is masked, 2) simulations can guide experiments with respect to power, but consideration of false positive rates would be equally useful to empirical biologists. If the authors consider these points to have merit, then incorporating those ideas into the paper would be helpful.

Regarding issue 1), we are grateful that you identified this potential issue. To better clarify in the manuscript, we address this potential issue in a new paragraph (L.243+) that discusses experimental caveats. Specifically, we added the following to L247-252:

“Moreover, overly strong selective regimes (e.g., high mortality) may deplete genetic variation in pleiotropic alleles and mask subsequent ontogenetic pleiotropy. We encourage the use of pilot studies to delineate how much stress can be applied so that numbers (and genetic variation) are not overly depleted. In addition, we encourage the use of species with high fecundity and high genetic diversity to reduce issues of depleting genetic variation (See study system considerations).”

Regarding issue 2), the power analysis simulates neutral alleles with no allele frequency change and uses the quantiles on this distribution to test the null hypothesis. Therefore, we calculated power as the proportion of times the observed allele frequency change was outside the 95% quantiles of the null distribution. Because the analysis is based on permutation, the false positive rate is fixed at 5%. This information was in the supplemental methods but is now also included in the main document on L260-261.

My main comments are aimed at clarifications and presentation. This is one of the few papers that reads as overly short and in various places concrete biological examples are neglected. I wonder if this brevity is due to journal word restrictions – if that is the case, I would encourage

the journal editors to allow some latitude because the paper will be stronger with a slight increase in length. Specifically, the paper's topic would be more compelling if the ubiquity of discrete life stages across the tree of life was made obvious within the first few paragraphs.

To emphasize the ubiquity of discrete life stages while highlighting the diversity therein, we have inserted the following in the beginning of the manuscript (L62-64):

“Across Eukaryotes, there is a myriad of different life cycle types that can include complex morphological changes within a single ploidy stage (e.g., metamorphosis) or among ploidy stages (e.g., haplodiplontic life cycles)[1–4].”

Similarly, the paragraphs on the developmental constraint hypothesis, eco-evolutionary perspective, and population genetics presents as quite vague: adding some concrete biological examples would underscore the points being made and animate the paper. For instance: “...there is experimental support for this Ho for taxa x, y, and z and no support for r, s, t but this hypothesis has not been comprehensively evaluated for taxa with life cycles that include....”

The review section of our paper was also the subject of discussion from the other reviewers who identified a high degree of conceptual overlap with another paper. In response to both reviews, we have decided to substantially trim the “review” portion of this section and direct readers to Collet and Fellous 2019 who provide a more thorough review. Instead, we now provide just the conceptual information necessary to justify and understand the goals of the manuscript. As a result of this shift in perspective, providing specific examples is no longer relevant to this section. This section extends from L88-144.

The supplemental should supply more information on the literature survey results – full species names should be provided (“marine invertebrate” is extremely very vague). Reporting the DOI is fine, but a full citation should be provided too. And another remark aimed to the journal: the journal should figure out a way to provide citation “credits” to original data producers.

The requested information has been added to the supplementary table on the literature survey. Supplementary Table 1 now includes an additional column with the full reference. Whenever possible, full species names were added.

In summary: I really like this paper and my comments are suggestions aimed to strengthen its relevancy to empirical biologists. The journal could consider relaxing strict word limits if required and should credit all the papers that have informed this manuscript.

Minor comments:

100 consider explicitly defining pleiotropy at first mention because different readers may have subtly different mental models for this term

We now briefly define pleiotropy at first mention (L86) with a more developed definition in L154-155) and include a brief discussion on the difference between pleiotropy and genetic correlations in Box 1 (glossary).

100-101 missing word?

We have removed this sentence in the process of streamlining the section.

117, 130 Are pleiotropy and adaptive decoupling different points along a continuum (extent to which a loci has multiple phenotypic outcomes) or are they fundamentally different? Clarify.

We understand pleiotropy to be the result of genetic correlations, but not all genetic correlations are pleiotropic. For example, genetic correlations can also occur from alleles at linked loci (linkage disequilibrium). Thus, it is possible that some alleles can produce correlations that are not considered pleiotropic. However, if loci or alleles are known to be involved in pleiotropy across stages, there should be some degree of genetic correlation across stages, making complete decoupling unlikely. Therefore, while these concepts are related, they are not necessarily two ends of the same spectrum. We hope to improve clarity around this nuanced topic by including a brief discussion on pleiotropy and genetic correlations in Box 1 (glossary).

136 I agree with this statement but it would have been nice early on to establish the prevalence of this phenomenon with biological examples.

This is no longer applicable given the rewrite of this section.

150 -151 This clarity is what I was looking for in the previous section!

153 perfect clear definition

189 follow up with a definition/ example of the evolve and re-sequence experimental design for readers that might not be familiar with that construct

We define evolve-and-resequence L193 with the following:

“Evolve-and-resequence experiments, where selection is applied across multiple generations and paired with iterative genomic sequencing, are commonly used to test for selection responses across generations [34]”

192 follow up with an example of what an across life stage experiment might look like

We provide a visual depiction of the experimental design as well as hypothesized outcomes in Figure 1.

Fig B1 - Although I like this figure conceptually, some modifications will highlight the nuance a bit more. For the marine example, depicting the duration and habitat used for each stage will increase accuracy and (with correct habitat identified) underscore the sometimes profound differences in selective challenges faced by different stages. The kelp background, while attractive, implies that all development occurs there, which is not true. Highly disparate stages are probably more likely to incur trade-offs. Also, in keeping with the spirit of the paper's intent to unify thinking across environments, a terrestrial example would be a welcome complement to the kelp forest example.

We agree that the kelp background oversimplified the where and when these different stages occur. After several unsuccessful attempts to redesign the figure, we decided to retain the kelp image and ecosystem but edited the legend to indicate that this is a stylized graphic representing a single, well-known ecosystem and highlight that not all life cycles occur within this single biome.

The new figure legend now reads:

“Figure B1. To illustrate a range of diversity found in a single, well-known ecosystem, we show the life cycles of a sea otter, an urchin, a crab, and a kelp, though we note that not all life cycle stages occupy the kelp forest habitat (image background) exclusively. Different shapes refer to different life cycle stages: larval or spore (circle), additional larval or early development stage (triangle), juvenile (trapezoid), and adult (square/rectangle). Diplontic life cycles are shown here by a sea otter, urchin, and kelp crab. Haplontic life cycles (not shown here but can be found in dinoflagellate lineages) will be most like the otter life cycle as there is only a free-living haploid stage. Haplodiplontic life cycles, shown by the giant kelp, have dispersive zoospores, early developmental sporophytes (triangle), and juvenile stages that lead to the adult sporophyte (shown in center; we do not show juvenile gametophytes). Gametophytes (far right square) are microscopic. Background image is of an *Eisenia (Ecklonia) arborea* kelp forest at Santa Catalina Island (photo credit: SA Krueger-Hadfield) and line drawings were provided by Kathryn Schoenrock.”

Fig 1 - I do not understand why you expect fitness in the experimental to decrease at Stage 2 under ontogenetic decoupling; shouldn't the fitness of experimental and control look the same here (slope = 0)? Similarly under synOP I would expect fitness in experimental to be higher

(conditioned on the S1->S2 transition). The figure legend restates the expectation shown in the figure but does not explain why.

The experimental treatment is an environmental stress. Under ontogenetic decoupling, the focal allele confers no benefit in stage 2. The increase in frequency of the focal allele from stage 1 to 2 in the experimental treatment thus confers no benefit to stage 2 and thus the experimental treatment again reduces fitness (so the fitness of the experimental treatment is lower than the control). This is assuming the experimental treatment confers the same fitness reduction on stage 1 and stage 2 of this species.

Under synOP, the fitness of the experimental relative to the control treatment depends on the degree of benefit of the focal allele. Here, we were assuming the focal allele increases fitness enough to bring the experimental treatment close to the fitness of the control treatment (reduces the net negative benefit), but not enough to make the fitness in the experimental treatment higher than the fitness in the control treatment.

We have redesigned Fig. 1B to make it easier to interpret and clarified these issues in the figure caption. Specifically, we write:

“Figure 1: Depiction of a general framework (A) and expectations (B) when testing for changes in allele frequency in response to selection across life stages within a single generation. ΔP is the change in allele frequency between stages. Different shapes indicate different morphologies of a species as it passes through the different phases of its complex life cycle (Box 1 and Fig. B1). In Stage 1, individuals are exposed to control or experimental treatments. For the purposes of this diagram, we assume the experimental treatment reduces survival (fitness) to the same extent in Stages 1 and 2 but the focal allele increases survival in Stage 1 and potentially in Stage 2 depending on the hypothesis. Thus, the focal allele increases in frequency in the experimental treatment in Stage 1 for all hypotheses. Both groups are reared through Stage 2 in the same treatments, with fitness and allele frequencies remeasured. The control treatment is not subject to a selective pressure, so the survival (fitness) and allele frequencies do not change under any hypothesis.

Is there a concern that if overly strong selection is induced at S1 that variation is depleted and therefore any OP response could be masked?

We have included a new paragraph that addresses this concern and provide potential solutions beginning in L243.

Overall I quite like this figure and appreciate that it includes both a concept diagram and pseudo data expectations.

Thank you very much for your thoughtful review of our work.

Cynthia Riginos

Referee: 2

In this paper the authors address the issue that current understanding about the ecological and evolutionary constraints imposed by life cycle complexity is limited, despite that complex life cycles abound in nature. This is an important issue and a topic that definitely deserves more study. The authors discuss genomic approaches to study selection across the multiple stages that make up a complex life cycle and propose a framework for hypothesis testing. Despite that the topic is relevant, important and timely I cannot recommend publication of this manuscript for the following reasons.

We would like to thank this reviewer for their careful consideration of our manuscript and the thorough review.

- Most importantly, in my opinion there is significant overlap with a recent article in the Proceedings of the Royal Society on exactly the same topic (Collet J, Fellous S. 2019 Do traits separated by metamorphosis evolve independently? Concepts and methods. Proc. R. Soc. B 286: 20190445. <http://dx.doi.org/10.1098/rspb.2019.0445>). The authors might have missed this paper as they do not refer to it, but I do not see how the current manuscript advances on this earlier publication.

The Collet and Fellous 2019 paper provides an excellent and thorough review of concepts and methods used to study evolution across metamorphosis. As a result, we have largely shortened the “overview” section and now refer readers to the Collet and Fellous paper (L 80; L 125) which provides a more complete review of these concepts, though we retain information necessary to understand and interpret our own manuscript. In addition, we have made significant efforts to clarify important differences between our papers. For instance, we have added the following sections to our paper:

L78-87: “Our goal is to provide a generalized conceptual and novel experimental framework that can be used broadly across eukaryotes and importantly, is not restricted by taxon or by the specific details of the life cycle (Box 1). These goals build upon [12] who considered similar questions from a quantitative genomic perspective. However, the maturation of genomic methods has generated powerful new opportunities to understand how selection acting within individual life stages affects adaptive potential in response to changing environments. Here, we explore the value of genomic analysis of single-generation artificial selection experiments

because they can provide direct evidence of genetic correlations (pleiotropy) in terms of allele frequencies, potentially minimizing the confounding ambiguities generated from environmental effects.”

L123-125: “For a full review on concepts related to adaptive decoupling, genetic correlations (with respect to traits), pleiotropy, as well as quantitative genomic methods traditionally used to study these concepts, we refer the reader to [12].”

L126-144: “Quantitative genetics has been used to address our understanding of the diversity of complex life cycles across eukaryotes in the past [12]. Moreover, empirical work on genetic correlations across stages has predominantly focused on one or two stages, with few studies quantifying selection across the entire life cycle (see Box 2). In these studies, the response is usually measured as a phenotypic trait at discrete life stages that affects fitness. But the true fitness value of a trait also depends on how the expression (or lack of expression) of that trait in subsequent stages affects performance, so isolating fitness metrics within single stages can give a misleading impression of overall adaptive potential [8]. We present a new conceptual and experimental framework that bridges the evo-devo and eco-evo viewpoints with a population genetic perspective to form a groundwork upon which we can build a more complete understanding of how stage-specific selection may promote or constrain adaptation to environmental change. We present a genomic framework because many species do not have traits that are directly comparable from one life stage to the next (e.g., oral arm on an urchin pluteus larvae vs. spine length on an adult urchin), whereas an allele is a unit that is directly comparable from one life stage to the next. Second, many species have traits relevant to climate change that are difficult to measure at the phenotypic level but easier to measure at the genomic level among life stages. For example, thermal tolerance between a kelp gametophyte vs. sporophyte would be difficult to measure given little phenotypic overlap, whereas expression of heat shock proteins would be directly comparable.”

L237-239: “Our design specifically uses environmental treatments to look at the genetic effect of stressors, contrasting with the approach of [12], in which environmental effects and plasticity are controlled to reveal genetic effects.”

- One important issue that I have with the current manuscript is that the authors use “alleles” and “traits” interchangeably. This lack of a clear distinction between alleles and traits results in confusing statements. Consider as one specific example line 171-174: “*Ontogenetic decoupling, AntOP, and SynOP are likely operating on different alleles and/or traits within the same genome. The distribution of the three categories across the genome is an area of research deserving increased attention, as well as how the nature and distribution of the three categories changes as the environment changes*”. The “*distribution of traits within the same genome*” only makes sense if alleles are equated to traits.

We have improved clarity by replacing these terms throughout the manuscript with the more general, “fitness components”. We define fitness components in line XXX 139 with the following:

L93-95: “Fitness component” refers to any life history trait or allele that is correlated with within-stage fitness or total fitness when all other traits or alleles (respectively) are held steady [16].”

And further delineate meaning with the following in L148-151: “When a fitness component (allele or its resulting trait value) that affects within-stage fitness at one stage is neutral at other stages (i.e., does not affect within-stage fitness), the impact of selection is decoupled because the effects are isolated to a single life stage”

And in L156-158: “We expand this to situations when an allele and resulting trait value expressed at one stage also affects that trait or another trait expressed at other life stages and refer to this as “ontogenetic pleiotropy”.”

We also note that we are not using the phrase “distribution of traits within the same genome” at any point in the manuscript.

- In contrast, in their review Collet & Fellous use traits throughout their presentation, while allowing for genetic correlation between traits in different life history stages due to pleiotropy. I consider this latter conceptualization a much more logical approach. It implies, however, that genomic approaches are much less useful for studying natural selection in the context of complex life cycles, in contrast to what is argued in this manuscript.

It is correct that the use of trait vs. our focus on genomics is an additional difference between our manuscript and the Collet and Fellous 2019 paper. While Collet and Fellous focus on traits and on species that undergo metamorphosis, a primary goal of our paper (as indicated in L78-82, L89-99, and Box 1), is to provide a framework that is not limited by the biological level in focus (i.e., expands beyond a trait focus) or limited by the specific details of the life cycle (whether the organism undergoes metamorphosis or not). Another primary goal of our paper is to present a framework that shows that a genomic focus is useful (and in some cases necessary) for studying selection across life stages. We now include the following text in L138-145 to emphasize this point:

“We present a genomic framework because many species do not have traits that are directly comparable from one life stage to the next (e.g., oral arm on an urchin pluteus larvae vs. spine length on an adult urchin), whereas an allele is a unit that is directly comparable from one life stage to the next. Second, many species have traits relevant to climate change that are difficult to measure at the phenotypic level but easier to measure at the genomic level among life stages. For

example, thermal tolerance between a kelp gametophyte vs. sporophyte would be difficult to measure given little phenotypic overlap, whereas expression of heat shock proteins would be directly comparable.”

- Another major problem is that the authors talk about fitness but never clearly define it. The authors start off by adopting the entire life cycle as the central unit, which makes “fitness” something that can only be assessed over the entire life cycle (as it should be). But in their conceptual framework subsequently discuss make statements like: “*a single allele or trait that affects fitness at one life stage also affects fitness at another life stage*”. In fact, the 3 “hypotheses” as the authors call them of ontogenetic decoupling, antagonistic ontogenetic pleiotropy, and synergistic ontogenetic pleiotropy rely on this “fitness-within-a-stage” concept and hence contrasts with the initial position that the entire life cycle is the central unit.

We now provide clearer definitions and use these terms throughout the manuscript to improve clarity surrounding fitness. Definitions are provided in L95-97 that reads:

“Within-stage fitness” describes the effect of fitness components on survival or reproduction within an isolated stage, while “total fitness” describes the cumulative effect of fitness components on survival or reproduction across all life stages [17].“

- In a similar vein, the evolve and resequence approach applied within a single generation that the authors propagate as a method of study strikes me at odds with the tenet that fitness is a characteristic of the entire life cycle. This approach only makes sense if the analysis is restricted to individual survival, as the authors do in Figure 1.

We have improved the specificity of our language (e.g., distinguishing within-stage fitness and total fitness), and we added the following to L205-208:

“Nonetheless, allele frequencies and phenotypic trait means are metrics that can indicate a response to selection for fitness components within each stage or inform the potential for adaptive responses when total fitness (across the whole life cycle) is measured (See Supplemental Table 2).”

Finally, we added an additional table (Supplemental Table 2) that gives example scenarios that would indicate responses to selection for fitness components that affect evolutionary outcomes. We hope these changes clarifies the insights that can be provided by our experimental design.

In contrast to Collet & Fellous who consider gene x environment interactions, the authors pay almost no attention to an individual’s environment. Their focus is almost exclusively

on genes, alleles and genomic approaches. I struggle with that lack of consideration for the environment.

Our design uses environmental treatments (stressors) to identify relationships across stages and how these may be changed in a changed environment. However, we now include a new paragraph that provides further discussion on the experimental design to include more detail on the environmental effects. This section also further underlines another difference between our approach and the Collet and Fellous paper. This paragraph (L233-143) reads:

“We use this simple design to provide a baseline of the experimental requirements needed to effectively study ontogenetic decoupling, AntOP, SynOP, and how these relationships affect responses to climate change. However, it is worth noting that this design can easily be modified to allow for more complex designs (such as fully crossed, factorial designs). Our design specifically uses environmental treatments to look at the genetic effect of stressors, contrasting with the approach of [12], in which environmental effects and plasticity are controlled to reveal genetic effects. If environmental effects or gene-by-environment interactions are of interest to researchers, this experimental design can be expanded from the current two-environment design to an environmental gradient/dose-response design that improves resolution of environmental effects on ontogenetic decoupling, AntOP, SynOP across life stages.”

- As a consequence of the previous point, the idea of “ontogenetic decoupling” in my opinion only makes sense in a context where there is absolutely no interaction between individuals. Reduced survival in one particular stage, for example, translates into fewer survivors in the next stages, which means lower numbers of competitors and/or fewer partners for mating. In other words, in a population context different life stages are always coupled demographically, which only has no bearing on fitness in the rather extreme situation in which individuals do not interact with their conspecifics. This demographic coupling inherent in complex life cycle is ignored by the authors.

We agree that density can be a major confounding effect. To deal with this (and other potential issues identified by other reviewers) we have added the following paragraph with important considerations to make in the application of the experimental design in L244-257:

“Issues of density, developmental rates, and genetic diversity are important considerations in the implementation of this design [38]. For example, when environmental stressors cause differential mortality across pools of individuals, density-dependent changes to developmental rates may be expected to confound experimental results if not accounted for. Moreover, overly strong selective regimes (e.g., high mortality) may deplete genetic variation in pleiotropic alleles and mask subsequent ontogenetic pleiotropy. We encourage the use of pilot studies to delineate how much stress can be applied so that numbers (and genetic variation) are not overly depleted. In addition, we encourage the use of species with high fecundity and high genetic diversity to reduce issues of depleting genetic variation (See study system considerations). To control for

different developmental rates, we suggest collecting additional samples that account for both time and developmental stage. We acknowledge that these solutions complicate the experimental design and remain an issue for selection studies that warrants broader discussion. Nonetheless, these solutions can preserve the ability to gain insights from short-term experimental designs.”

- Lastly, the authors do not really distinguish between life cycles with or without metamorphosis, as illustrated by their Figure B1 which contains both giant kelp, sea urchins and sea otters. In many species the complex life cycle also involves metamorphosis, which is a very costly, complete overhaul of an individual’s body plan. This component of the fitness of an individual life history is totally ignored in this manuscript.

Our approach expands beyond the focus on metamorphosis and remains intentionally agnostic to the complexity of the life cycle (which we specify in L78-80). We agree that latent effects and carryover effects could be important, which has arguably dominated our understanding of complex life cycles. While we agree that metamorphosis can be a very costly process for some taxa, and that latent effects and carryover effects may affect later outcomes, the goal of the paper is to understand the cumulative effect across transitions for any life cycle.

Signed,

André de Roos

Referee: 3

Comments to the Author(s)

In this review, Albrecker et al. synthesize the importance of integrating the effects of selection across life cycles when considering adaptation to environmental change. They then propose an experimental framework for using genomics to test how the fitness effects of alleles vary across stages and conduct simulations to provide guidelines for appropriate sample sizes and sequencing coverage.

Considering selection across the entire life cycle is an extremely important and often overlooked component of adaptation. Additionally, I think the idea of applying evolve and resequence thinking to within-generation selection is quite clever. I expect that this paper could be used a framework for many studies looking to understand the selection across complex life cycles.

While I think the concept is extremely interesting and important, I did feel that there were large holes in the ‘review’ component of the paper and the text could be generally streamlined. I

understand the authors may be space limited, but I think that the background and box sections could be used to provide researchers more context for what we already know about trait correlations across metamorphic boundaries.

Thank you for your review including many very helpful comments and considerations that helped streamline and improve our manuscript.

Although the following are purely my opinion, here are some specific ideas for what I felt could be added/removed:

1) For me, the presentation of the separate fields (evo-devo, eco-evo, population genetics) that are meant to be synthesized did not add much to the paper. In fact, I felt some of the most important contributions of each were overlooked. For example, the field of evo-devo has elucidated the function of many important genes. Also, the paragraph on population genetics reads more like a paragraph on quantitative genetics, highlighting correlations in trait expression.

We have largely re-written and streamlined this section of the manuscript in response to comments from all three reviewers. Specifically, we no longer attempt to “review” the field but rather provide only information necessary to justify the goals, hypotheses, and experimental designs in this manuscript. We refer the reviewer to the new overview section beginning in L88.

2) While I appreciate that the authors are trying to highlight the diversity of life cycles and clarify some terminology, I felt that Box 1 was largely tangential to the main point of the paper.

While we appreciate the reviewer’s concern that box 1 is tangential, we suggest that this comment illustrates one of the main goals of our paper. In general, the field has overwhelmingly focused on a very narrow swath of eukaryotic diversity. Moreover, terminology across different taxa and systems can be misinterpreted - for example, life history and life cycle are often used as synonyms when they are in fact not. We hope to increase appreciation of eukaryotic life cycle diversity by developing methods that can be applied to more than invertebrates with metamorphosis. While work on adaptive decoupling in invertebrates has no doubt greatly expanded our understanding of eukaryotic evolution, to continue in such a myopic approach does not enable us to understand how other taxa with complex life cycles will respond to climate change. We have added text on L78-80 to emphasize these points including:

“Our goal is to provide a generalized conceptual and novel experimental framework that can be used broadly across eukaryotes and importantly, is not restricted by taxon or by the specific details of the life cycle (Box 1).”

3) In place of some of the above content, I would have liked more of a true review of correlations in traits across metamorphosis. This is super important for establishing the need to understand the variation selective affects. I've listed a few examples at the end of this review.

Collet and Fellous 2019 (*Proceedings B*) provide an excellent and thorough review of this topic. Given the conceptual overlap, we now refer readers interested in a full review to that paper, and instead only retain information necessary to interpret our manuscript (L 80; L125-126). The goal of our manuscript is to motivate researchers working on eukaryotes with complex life cycles to consider genomic approaches to quantify how environmental change will affect selection across stages of a life cycle. We outline an experimental framework that can be adapted to a variety of species.

4) Similar to above, the authors are not the first to consider pleiotropy across developmental stages (see for example Donohue 2014) and I would like to see more previous work discussed. Highlighting what we know about this field is again important to the main argument of the paper.

See response to comment 3. We recognize that these concepts are not new, but our conceptual framework, hypotheses, and experimental framework are novel contributions to the field. As indicated previously, it is our hope that our proposed genomic framework will allow for a greater degree of understanding and communication across study systems and biological subdisciplines.

For the experimental design, I'm concerned about the lack of a proper control for developmental plasticity. You can imagine, for example, that epigenetic (or other) effects before stage 1 might alter the fitness consequences of particular alleles. Although this may be 'realistic' if organisms are in a constant environment across their life cycles, it is less so in fluctuating environments. To get a more 'true' measure of the fitness affects of each allele at each stage, it seems like you would need a fully crossed design.

We now include a new paragraph that suggests modifications to the experimental design depending on the researcher's goals. It begins L233 and reads:

“We use this simple design to provide a baseline of the experimental requirements needed to effectively study ontogenetic decoupling, AntOP, SynOP, and how these relationships affect responses to climate change. However, it is worth noting that this design can easily be modified to allow for more complex designs (such as fully crossed, factorial designs). Our design specifically uses environmental treatments to look at the genetic effect of stressors, contrasting with the approach of [12], in which environmental effects and plasticity are controlled to reveal genetic effects. If environmental effects (plasticity) or gene-by-environment interactions are of interest to researchers, this experimental design can be expanded from the current two-

environment design to an environmental gradient/dose-response design that improves resolution of environmental effects on ontogenetic decoupling, AntOP, SynOP across life stages. “

Appendix C

Dear Dr. Trussell and editors:

We are thrilled that our paper has been accepted pending minor revisions. We are very grateful for the kind and helpful reviews that have substantially improved our work. We have made the minor revisions as requested and highlight these changes in blue below.

Kind regards,

Dr. Molly Albecker and Dr. Laetitia Wilkins on behalf of the authors

Dear Dr. Albecker,

I am pleased to report that I have received two reviews of your revised manuscript, "Does a complex life cycle affect adaptation to environmental change? Genome-informed approaches for characterizing selection across life cycle stages". Both reviewers agreed that you did a nice job addressing comments on your first submission and recommend acceptance of your paper for publication. I concur. Please note that one reviewer has some minor suggestions that you may want to consider as you prepare your final manuscript.

Congratulations on developing a nice paper and I look forward to seeing it "in print"!

Cheers,

Geoff

Thank you, Geoff!

Reviewer(s)' Comments to Author:

Referee: 1

Comments to the Author(s).

The revised version of the manuscript has been streamlined in focus and has referred most biological examples to a recent review paper. This solution works fairly well and is helpful for maintaining a shortish format. Since the focus of the present manuscript now shifts more heavily towards "evolve and resequence" experiments, a bit of review (what sort of organisms, what sorts of experimental designs) would be welcome if there is room and appetite.

We agree that a review of this topic would be interesting, however we opted not to include it here given space constraints and the fact that would not directly relate to the goals of the manuscript.

Any substantive points that I had previously raised have been appropriately addressed. A few minor points follow, aimed at improving clarity and relevancy for future readers.

Lines 130-134 I appreciate that you have slimmed down the review aspects of this manuscript. Nonetheless, in this paragraph a concrete example of a noteworthy study would really help make the abstract ideas more tangible. Just a single sentence that highlights one example where discrete life stages were examined would give a sense how experiments are done and what can be found.

Thank you! We have added the following example to the text in lines 132-136:

“For example, although size is commonly used as a fitness correlate for amphibians as they undergo metamorphosis, compensatory growth in subsequent life stages can eliminate or reverse size differences among groups, which may limit the overall impact of metamorphic size on the lifetime fitness of the individual.”

139-140 Very helpful examples!

T1: Give full names to SynOP and AntOP within T1 for readers who are skimming.
Changed as requested.

Fig 1 B - why are the Y axes have coloured and hashed lines with arrowheads? This is really confusing. Also, I think your labels for the Y axes are not quite accurate. Green/fitness appears to be relative survival from previous life stage - F(S2) makes no sense for a measurement taken at S1. Similarly, blue/allele frequency is not P(S2) but P(at each life stage).
Thank you for spotting this! This was an accidental hold-over from a previous iteration of this plot. It is now fixed along with the suggested changes to the axes.

Please proofread and correct your literature cites.
We have proofread the literature cited for obvious mistakes.

Referee: 3

Comments to the Author(s).
The authors have done a terrific job responding to reviews and I believe the paper is much improved. I have no further concerns.
Thank you very much!